# Development of Tools for Coastal Management in Google Earth Engine: Uncertainty Bathtub Model and Bruun Rule

Lucas Terres de Lima [1,*], Sandra Fernández-Fernández [2], João Francisco Gonçalves [3,4], Luiz Magalhães Filho [5] and Cristina Bernardes [1]

1 CESAM—Centre for Environmental and Marine Studies, Department of Geoscience, University of Aveiro, Campus de Santiago, 3810-193 Aveiro, Portugal; cbernardes@ua.pt
2 CESAM—Centre for Environmental and Marine Studies, Department of Physics, University of Aveiro, Campus de Santiago, 3810-193 Aveiro, Portugal; sandrafernandez@ua.pt
3 CIBIO-InBIO, Research Center in Biodiversity and Genetic Resources, University of Porto, Campus de Vairão, Rua Padre Armando Quintas, 4485-661 Vairão, Portugal; joao.goncalves@cibio.up.pt
4 Instituto Politécnico de Viana do Castelo, Rua Escola Industrial e Comercial Nun'Álvares, 4900-347 Viana do Castelo, Portugal
5 CESAM—Centre for Environmental and Marine Studies, Department of Environment and Planning, University of Aveiro, Campus de Santiago, 3810-193 Aveiro, Portugal; luizlacerda@ua.pt
* Correspondence: lucasterres@ua.pt

**Abstract:** Sea-level rise is a problem increasingly affecting coastal areas worldwide. The existence of free and open-source models to estimate the sea-level impact can contribute to improve coastal management. This study aims to develop and validate two different models to predict the sea-level rise impact supported by Google Earth Engine (GEE)—a cloud-based platform for planetary-scale environmental data analysis. The first model is a Bathtub Model based on the uncertainty of projections of the sea-level rise impact module of TerrSet—Geospatial Monitoring and Modeling System software. The validation process performed in the Rio Grande do Sul coastal plain (S Brazil) resulted in correlations from 0.75 to 1.00. The second model uses the Bruun rule formula implemented in GEE and can determine the coastline retreat of a profile by creatting a simple vector line from topo-bathymetric data. The model shows a very high correlation (0.97) with a classical Bruun rule study performed in the Aveiro coast (NW Portugal). Therefore, the achieved results disclose that the GEE platform is suitable to perform these analysis. The models developed have been openly shared, enabling the continuous improvement of the code by the scientific community.

**Keywords:** sea-level rise; geographical information system; open-ource software; modeling

## 1. Introduction

Coastal areas are densely populated gathering more than 30% of the world's population and this is increasing exponentially [1,2]. Depending on the geologic, climatic, and oceanographic conditions, coastal zones may present a high risk of flooding, erosion, and regional sea-level fluctuations among others. All these phenomena are natural and contributed to model present-day coastlines. However, in the last seventy years, the effect of these drivers has increased in both intensity and frequency [3,4], and it is expected that this rising trend keeps up in the future, being the anthropogenic forcing the main reason for the global average sea level rise since 1970 [5].

The sea-level rise is a common problem that affects about 70% of coastal zones worldwide that it is accelerating and it is expected to be worst in the future [6,7]. The global mean sea-level (GMSL) rose 0.16 m between 1902 and 2015. However, in the period 2006–2015, the GMSL rise rate was 3.6 mm yr$^{-1}$, about 2.5 times higher than in the period 1902–1990 (1.4 mm yr$^{-1}$). The ice sheet and glacier contributions over the period of 2006–2015 were the most important sources of sea-level rise (1.8 mm yr$^{-1}$), exceeding the influence of the thermal expansion of ocean water (1.4 mm yr$^{-1}$) [7].

Considering the dimension and complexity of sea-level rise hazards, the use of Geographic Information Systems (GIS) to organize and to analyze the information produced about those issues is crucial to improve coastal management. Desktop GIS applications such as ArcGIS [8], gvSIG [9], Terraview [10], or QGIS [11] have traditionally been used in coastal management. However the exponential increase of Google Earth Engine (GEE) [12] in terms of available data, and capability to address a considerable volume of datasets with high spatial resolution has become GEE capable of connecting large-scale problems on coastal management in a new point of view.

The Google Earth Engine is a cloud-based platform that offers high-performance computing resources for processing geospatial data [12]. It provides access to an increasing amount of remotely obtained datasets through its application programming interfaces (API) for JavaScript and Python languages, which decrease the complexity of laborious desktop-based computations [13].

GEE use has been growing very fast in the last few years. Several applications were developed, such as MapBiomas [14], which provide a historical dataset of land use maps; CoastSat that allows extracting coastlines from Landsat and Sentinel images [15]; and the extraction of bathymetry from Sentinel 2 images [16]. One of the benefits of creating models on GEE is the possibility to work efficiently and quickly on a large scale. These advantages can be integrated into scripts (based on GEE API) by implementing modeling frameworks and creating new tools and analysis methodologies, which can improve new knowledge and its application.

The simple Bathtub method is a GIS technique that shows the areas below a specific elevation level as being flooded, like a bathtub or single value water surface [17]. Based on the former, the uncertainty Bathtub model (uBTM) [18] is a modified version of this technique that combines the uncertainty of sea-level projections and the vertical error of a digital elevation model (DEM). Based on the Terrset sea-level impact tool [19], the model defines the probability of the sea-level to flood a considered zone, using the level of uncertainty associated with the DEM and the sea-level rise projections.

The Brunn rule for GEE model (BRGM) [20] is based upon a formula created to estimate the retreat of sandy beaches coastline in response to sea-level changes [21]. The Bruun rule has some limitations, and its application requests precaution due to the simplicity of the formula; the equation does not include some essential variables such as extreme washover events, changes in sediment budget, and anthropic action. However, the formulation shows accurate results in the history of its application [22,23], and allows to obtain better results than those produced by modern models, such as the profile translation model (PTM) [24].

The main objective of this work is to explore the potential of GEE as support for two models—uBTM and BRGM—and its validation in the context of coastal management problems. The uBTM model uses the uncertainties of sea-level projections and the vertical digital elevation model error to create a coastal flooding scenario. The BRGM model is based on the Bruun rule equation that generates a tool capable of determining the coastline retreat in a coastal stretch.

## 2. Materials and Methods

### 2.1. Study Sites

The models were applied and validated using a morphological dataset of the southern Rio Grande do Sul, Brazil, and Aveiro region, Portugal, since both sites are sensitive areas to trigger events, e.g., storms and sea-level changes, and are well known by the authors, that possibility easily identify possible errors in the created models: uncertainty Bathtub model (uBTM), and Bruun rule for GEE model (BRGM).

The Rio Grande do Sul coastal plain (RSCP) in the south of Brazil was chosen to validate the uBTM (Figure 1a) because it is a large low-lying coast (200,000 km$^2$) that becomes ideal to test the robustness of the model on a large-scale, and with about 3 million people living in this area [25]. Therefore, a new sea-level flooding analysis can give support

for the policymakers of the region. The RSCP is characterized by an extensive NE-SW sandy barrier system of 620 km [26]. The coast is wave-dominated, and tides have a subordinated role in coastal hydrodynamics and present a mean amplitude of 0.5 m and a maximum of 1.2 m. The wave climate is dominated by two-wave propagation patterns, one composed of S-SE swell waves of higher amplitude and longer periods; the second comprising locally generated waves, with shorter periods and a predominant E-SE direction. Swell waves have a mean significant wave height of 1.5 m and periods of 12 s. Sea waves are characterized by a mean significant wave height of 1 m and a mean period of 8 s [27].

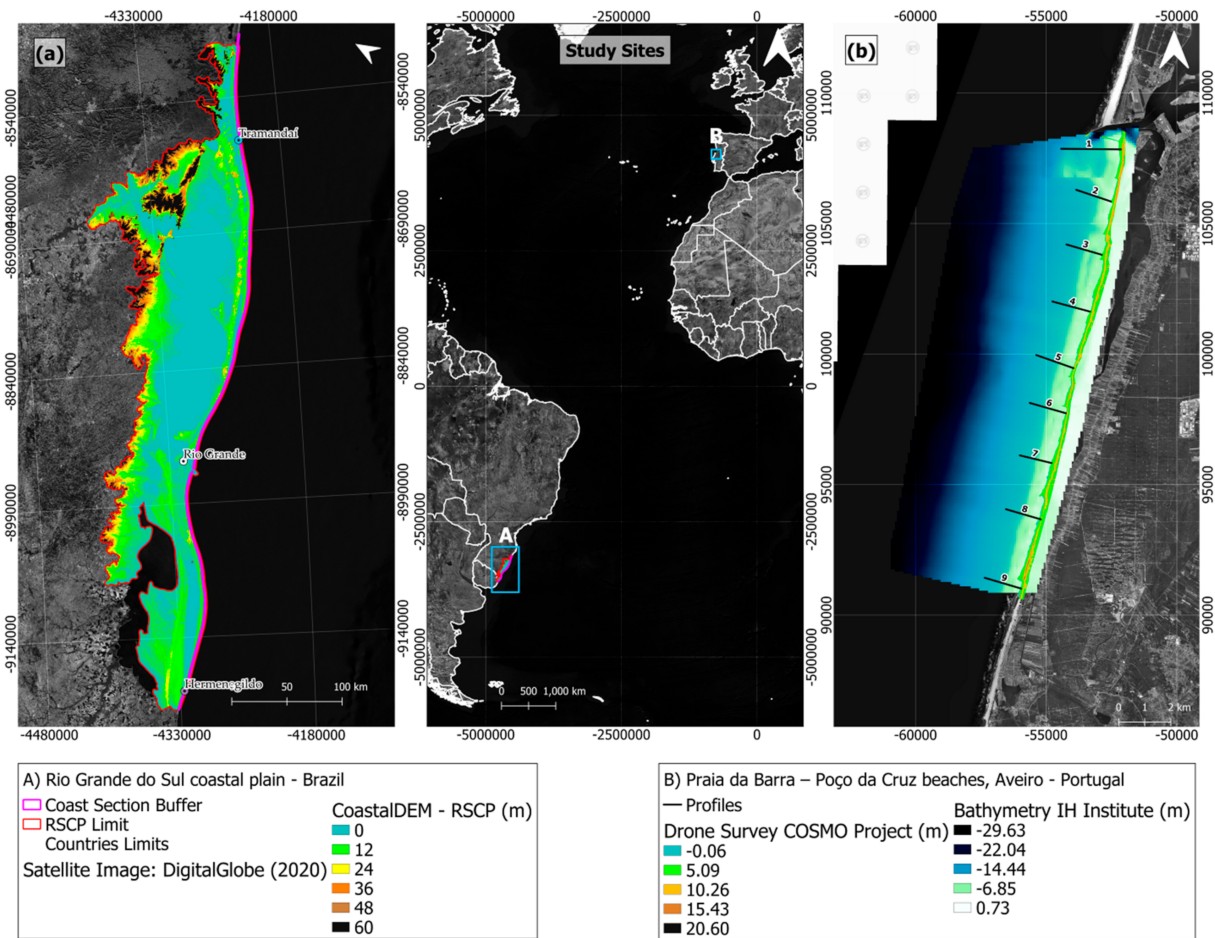

**Figure 1.** Study sites: (**a**) Rio Grande do Sul coastal plain, Brazil (coordinate system: SIRGAS 2000—UTM22). (**b**) South Aveiro lagoon entrance (Barra—Poço da Cruz coastal strecht), Portugal (coordinate system: ETRS 89 TM 06—Portugal).

The continental shelf is wide (100 to 200 km), shallow (100 to 140 m) and slightly sloping (0.02° to 0.08°) [26]. Differences in width, slope, and topographic features along the coastal region result from reworking action related to glacio-eustatic variations that occurred during the Quaternary [28]. The barrier system was formed in the last 7 Ka controlled by both sediment supply along the coast and morphology; coastal embayment promotes the development of regressive barriers and steeper coastal slopes are dominated by transgressive barriers [26].

Despite some erosional hotspots near the cities of Hermengildo [29], Rio Grande [30], Tramandaí [31], mainly due to human activities or extremes events occurrence, the coastline shows, in general, a stable or accretionary trend [29,32] (Figure 1a).

To validate the BRGM, the second site considered is Aveiro region, located on the northwest coast of Portugal (Figure 1b). The coastal erosion in this area is a problem, and there are important investments to protect the coastline and study the evolution of the coastline in the near future. The stretch is situated south of the Aveiro lagoon entrance

and is morphologically characterized by a sandy barrier extending in NNE–SSW direction. Nowadays, this area is highly vulnerable to erosion due to the very low and flat topography, combined with high wave conditions and a meso-tidal regime [33]. The sector considered, from Barra to Poço da Cruz beaches, is backed by a degraded foredune ridge partially destroyed by erosive processes and replaced by sand dykes. In general, the beaches show pronounced seasonal behavior, with a range of morphodynamical states. This variation reveals the important exchange of sediments between the upper and lower foreshore [34]. Despite this cross-shore transport, significant littoral drift causes major alongshore motion of sediments along the southwards direction [35]. However, the presence of several cross-shore structures (jetties and groins) contributes to changes in the sediment transport patterns.

The coast is exposed to highly energetic waves from WNW–NNW [36]. In maritime summer (June to September) significant wave heights and mean periods are less than 3 m and 8 s, respectively. During winter and transitions periods, the mean significant wave heights and periods exceed 3 m (most common values of 3–4 m) and 8 s (most frequent mean periods of 8–9 s), with storms defined by a mean significant wave height greater than 5 m (often exceeding 7 m) and mean wave periods of 13 s, which can reach maximum 18 s [37]. The average values for the spring and neap tidal ranges are 2.8 m and 1.2 m, respectively.

### 2.2. Uncertainty Bathtub Model

The model was entirely implemented on the GEE using the JavaScript API. The model analye the uncertainty of sea-level rise projections with vertical errors in the DEM, creating a frequency from 0 to 100%, which indicates the probability of a specific area to be affected by sea-level rise (Figure 2a). The model assumes the lowest vertical error of a DEM and the highest sea-level rise projection (Figure 2b). The areas that appear emerged are considered locals with a 0% of probability to be affected by sea-level rise flooding. On the other hand, a region has a 100% probability of being submerged when the maximum error of DEM elevation is compared with the lowest sea-level rise projection and the area appears submerged, even with optimistic settings (Figure 2c).

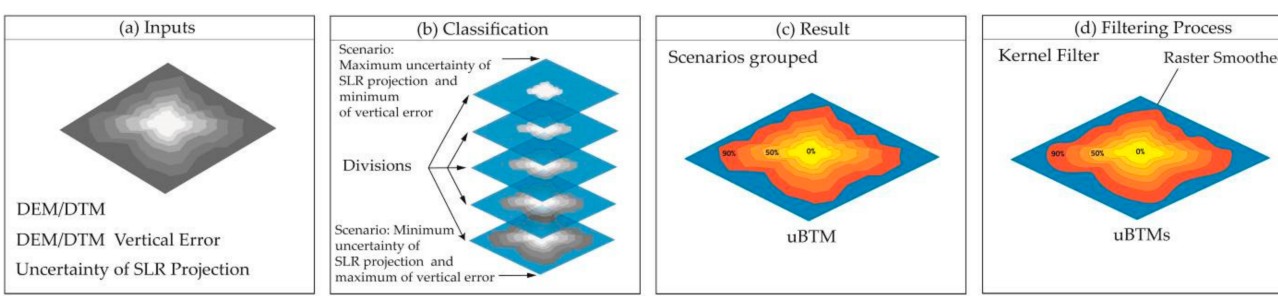

**Figure 2.** (**a**) Model inputs: Digital Elevation Model (DEM)/Digital Terrain Model (DTM) in raster format, vertical error of DEM/DTM, and uncertainty of sea-level rise (SLR). (**b**) ClasScheme 0. Named uncertainty Bathtub model (uBTM). (**c**) Result: all scenarios grouped into a single raster classified from 0% to 100% named uBTM. (**d**) Filter process (optional) is applied a Kernel filter on uBTM creating a smoothed uBTM (uBTMs).

The uBTM passed through a filtering process with different *Kernel Filters*, in order to choose the best option to smooth the data and to reduce both the pixelization and the image grain for a better delimitation of the waterline boundaries. All 3 × 3 *Kernel Filters* available on GEE (i.e., Cross, Plus, Gaussian, Diamond, Circle, Square, Octagon, Chebyshev, Euclidean, and Manhattan) were tested and compared. The geometric aspect of the circular *Kernel Filter* seems more adequate for the waterline shapes of the study area. However, this filter can be easily changed on the code and to be selected the one that is more appropriate to the coastal characteristics (for example, a square filter is reasonably proper in the case of

rocky cliff coasts). The result of the uBTM combined with the filtering process is called the uncertainty Bathtub Model smoothed (uBTMs) (Figure 2d).

### 2.3. uBTM Validation

The validation of the uBTM consists of performing a comparison between three similar GIS models (i) simple Bathtub model (sBTM), (ii) enhanced Bathtub model (eBTM), and (iii) Terrset sea-level impact (tSLI), which are briefly described below.

The sBTM is a user-defined static inundation water level that does not consider either the hydrological framework or physical barriers.

The eBTM includes a surface roughness coefficient that acts as critical variable influencing the water movement and the beach slope, to perform a more realistic representation of the area and coastal flooding conditions [38]. In the present case, the study region (RSCP) is mainly characterized by sandy substrate in the first meters above the surface [39–43] and according to [44] sands are characterized by a uniform roughness coefficient of 1.

The tSLI yields the effect of a sea-level rise integrating both the uncertainties of the projection and the DEM, using a *PCLASS* algorithm, which produces a probability image where values are between zero and one [19].

The calculation of the area by itself is not a good indicator of the similarity between the models because it ignores the spatial distribution. For this reason, a different methodology was developed allowing the quantification of spatial differences to check spatial similarities between models.

Several algorithms, including artificial intelligence, use heatmaps and statistical analyses to recognize objects and identify differences between images [45–47]. The method used to quantify the similarity of the spatial distribution consists of transforming the pixels values of the model into a density map and applying a correlation matrix to assess the similarities and their distribution. An ArcMap graphical model [8] was created to select only the impact using the *Kernel Density* tool to create a heatmap of pixel changes for each model (Appendix A). In the case of uBTM, uBTMs and tSLI—that show the impact from 0% to 100–50% is the point that expresses the value of sea-level on DEM without the influence of vertical error and uncertainties of sea-level projections. The use of the same values for sBTM and eBTM allows comparing both models.

It was necessary to remove the effect of lagoon areas (in the case of RSCP) and understand the spatial distribution near the coastline. For this reason, the same procedure to create heatmaps was performed by creating a small sector through a buffer area of 500 m from the coastline vector. The final step applied was the *Raster Correlations* and *Summary Statistics* of *SDM Toolbox v.2.4* [48], which creates a correlation matrix of the *Kernel Density*.

The accuracy of the method was verified by using two accessible *APIs* for image comparison: (i) DeepAI—image similarity [49] which uses an artificial neural network algorithm to identify the differences, (ii) *Resemble.js* based on the visual regression method [50]. The heatmap images of spatial distribution used for comparison were created by exporting from *ArcGIS 10.6* the *Kernel Density* images in a white background. In the end, the density points created on ArcGIS, the outcomes of the process of *DeepAI* and *Resemble.js*, and the results of the area differences calculation were correlated using *pyplot* library and *GoogleColab* [51]. *Matplotlib* is a library for producing visualizations (i.e., charts) in *Python* [52]. The *GoogleColab* platform allows writing and executing python code through the web browser in a cloud environment [53].

The DEM used to perform the analysis was the CoastalDEM free version (resolution of 90 m), a product created with a multilayer perceptron (MLP) artificial neural network to reduce the vertical error of shuttle radar topography missions (SRTM) to ca. 2.5 m [53].

The sea-level rise values were extracted from the regional data of special report on the ocean and cryosphere in a changing climate (SROCC) [5] under Representative Concentration Pathway-RCP 8.5. The value adopted is 0.68 m with uncertainties of 0.50 m to 0.90 m, for the period 2081–2100.

### 2.4. Bruun Rule implementation on Google Earth Engine

The Bruun formula [21] uses the berm height, the horizontal length after the berm ridge towards the backshore, or the beach face [54]. If the profile does not have a berm, the dune foot is considered. The code developed on GEE requires the sea-level rise projection, DEM (raster format) of topography, and bathymetry (Figure 3a) to create the topo-bathymetric profile, i.e., a line that allows obtaining the values of the berm height and depth of closure (Figure 3b). After that, Equation 1 was used with the values extracted from the created line. The displacement representation in the future is represented by using a simple buffer, with the extreme edge of the polygon being the final position of the coastline (Figure 3c).

$$R = S(W/h + B) \tag{1}$$

Bruun's classic equation where R is the coastline retreat, S is the predicted sea-level rise, W is the profile length; h is the depth of closure, and B the berm elevation.

In the last years, modifications to the original Bruun equation were proposed by [55] to incorporate the landwards transport, and [56], that included the contributions of the cross and longshore sedimentary processes and the sediment budget (Appendix B). These variables were included in the code, but its precision was not evaluated in this study.

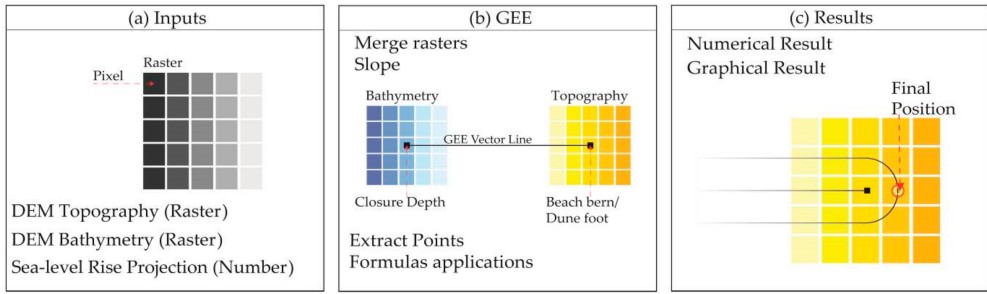

**Figure 3.** (**a**) Model inputs: DEM of topography and DEM of bathymetry, both in raster format, and the value number of sea-level rise projection. (**b**) Google Earth Engine: the rasters are merged and to calculate the slope. The depth of closure and beach berm values are extracted directly from GEE. (**c**) Numerical and graphical result, in the last case symbolized by a buffer.

In this study, the classic Bruun equation (Equation (1)) was applied on Barra-Poço da Cruz coastal stretch (Aveiro region), using bathymetry and UAV photogrammetry DEM data (Figure 1b). The results were compared with [57]'s study, which performed a Bruun rule analysis in the same region. Profiles for the GEE Model (BRGM) were created with the values detailed in this study. However, there are some bathymetric and topographic differences between the profiles performed by [57] and the present analysis because it was necessary to adapt the length of some profiles to get similar values of height and depth. The topographic and bathymetric data used have two sources, the COSMO Program [58] (topography) with 1 m of spatial resolution and the Portuguese Hydrographic Institute [59] (bathymetry) with 82.4 m of spatial resolution. Subsequently, a Spearman correlation analysis between the GEE Bruun rule results and the previous study [57] was performed. After the validation process, was tested for the area a scenario of sea-level rise of 1.21 m, considering the contribution of the Antarctic Ice Sheet (AIS) melting process (AIS = 60 cm) [60].

### 3. Results

The results section is divided in two subsections: (i) the validation process of the uncertainty Bathub model (uBTM) and (ii) Bruun rule validation for Google Earth Engine model (BRGM) through Spearman correlation analysis and the example of the application of the BRGM under a climate change scenario.

### 3.1. uBTM Validation

The subsection presents the results of the comparison of the areas between the different models, and those obtained with the spatial similarity using *Kernel Density* and machine learning APIs.

In Figure 4, the red color shows the areas with more than 50% of probability to be affected by a climate change scenario (RCP 8.5). The results are very similar in all the models performed, which means the outcomes of uBTM and uBTMs are coherent (Figure 4a,b). The regions in red color correspond to the total results of eBTM and sBTM (Figure 4d,e). Additionally, it is possible to observe the result of the *Kernel filter* smooth effect by using the uBTMs (Figure 4b, small frame). Besides, the Terrset Sea-level Impact (tSLI) showed the different spatial distribution of areas from 0 to 40% of impact (Figure 4c).

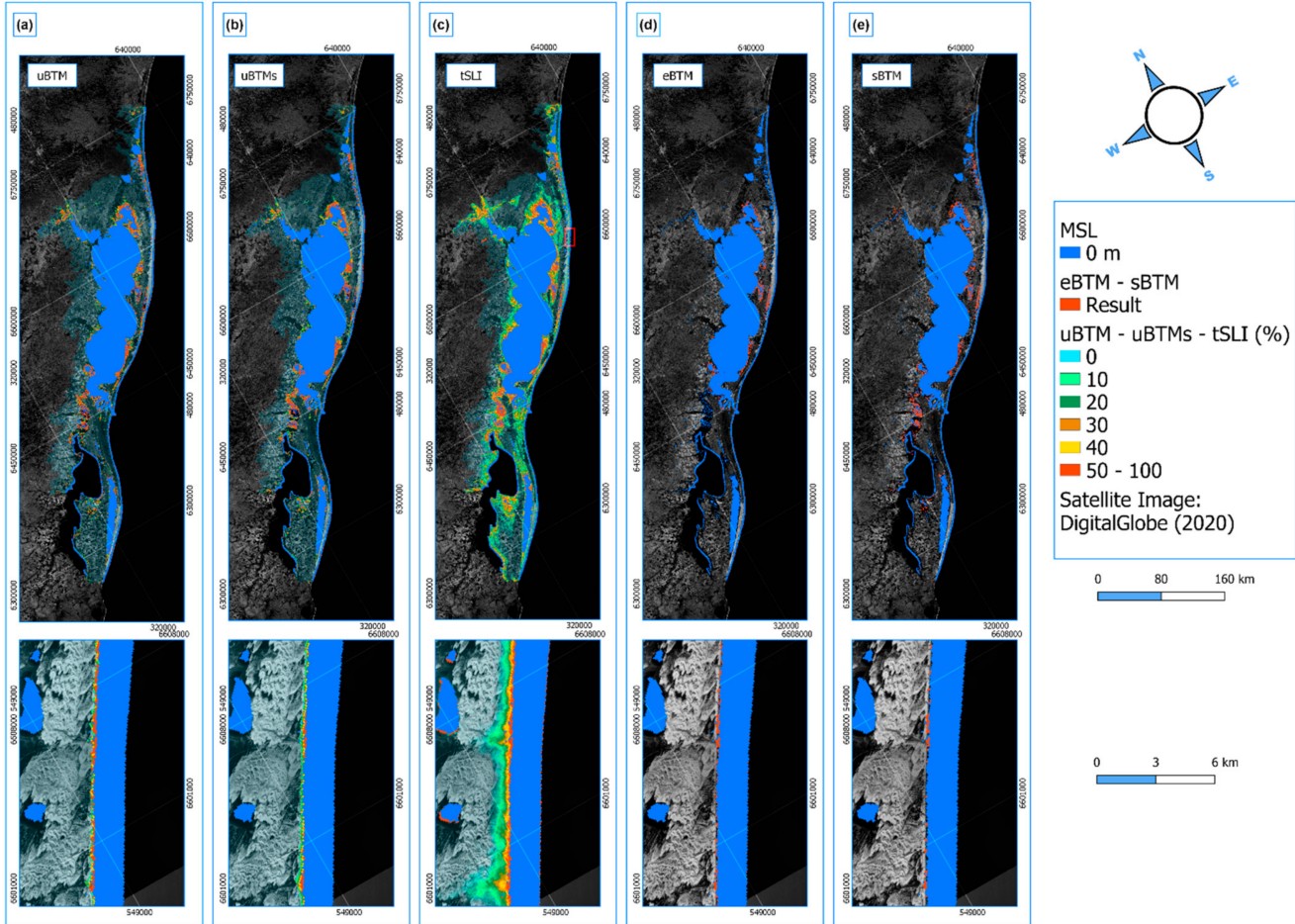

**Figure 4.** Rio Grande coastal plain impact results. (**a**) Uncertainty Bathtub model (uBTM). (**b**) Uncertainty Bathtub Model smoothed (uBTMs). (**c**) Terrset Sea-Level Impact (tSLI). (**d**) Enhanced Bathtub Model (eBTM) in red. (**e**) Simple Bathub Model (sBTM) in red.

The tSLI yields the most significant area affected, but the difference between tSLI and uBTM represents only 0.63% of the total area of Rio Grande do Sul coastal plain (RSCP) and 2.99% of the coastal stretch or section (CS). Furthermore, the total areas obtained by the uBTM and sBTM are similar (Figure 5). In the coast section, the eBTM, uBTM, and sBTM show also similar results.

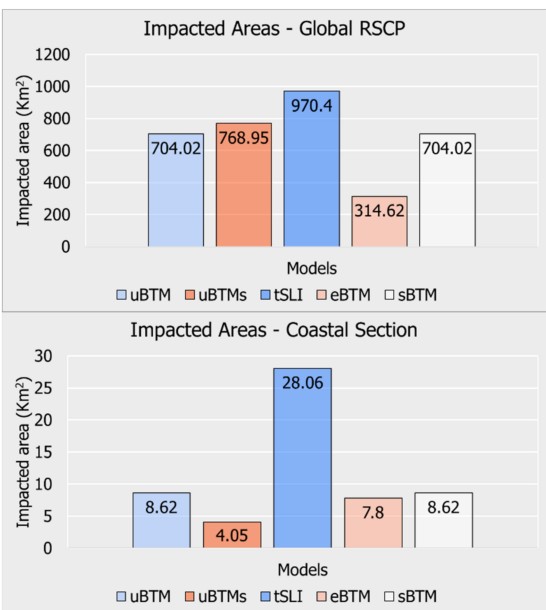

**Figure 5.** Calculated areas representing the results of the models on Rio Grande do Sul coastal area (RSCP) and coastal section (units in km²).

### 3.1.1. Spatial Similarity Analysis: Rio Grande do Sul Coastal Plain (RSCP)

The visual distribution of the impacts obtained by Kernel Density filter shows comparable patterns of clusters points regarding the lagoon margins and the coastline. Only in eBTM the spatial distribution is quite different due to the model characteristics. The hydrological features do not include the water bodies without connection to the ocean (Figure 6d).

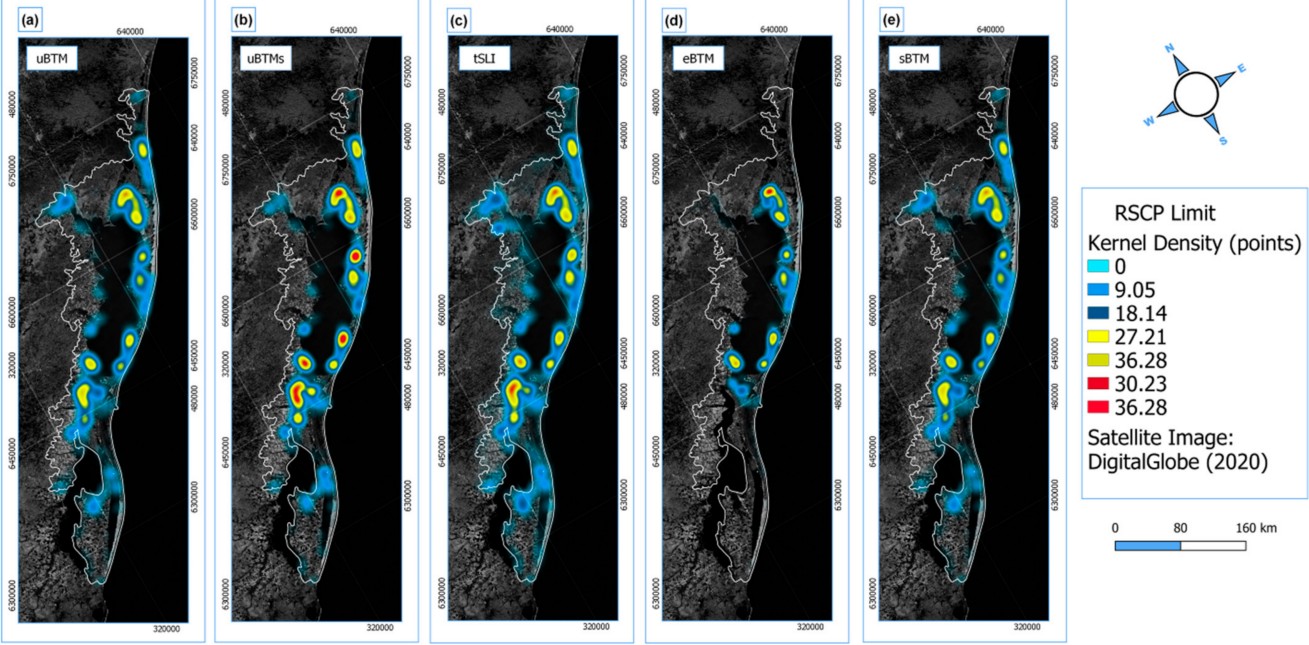

**Figure 6.** Spatial distribution through *Kernel Density* for the totally of Rio Grande do Sul coastal plain. (**a**) Uncertainty Bathtub model (uBTM). (**b**) Uncertainty Bathtub model smoothed (uBTMs). (**c**) Terrset sea-level impact (tSLI). (**d**) Enhanced Bathtub model (eBTM). (**e**) Simple Bathub model (sBTM).

The correlation matrix of *Kernel Density* results is presented in Table 1. The uBTM model shows a correlation of 0.99 and 1 with tSLI and sBTM, respectively. Moreover, the uBTMs display 0.97 of correlation with tSLI and sBTM. The low correlation values with eBTM were expected since the eBTM excluded water bodies that are not connected with the sea.

**Table 1.** Correlation matrix of models in Rio Grande do Sul coastal plain (RSCP).

|          | uBTM | uBTMs | tSLI | eBTM | sBTM |
|----------|------|-------|------|------|------|
| **uBTM**  | 1    | 0.97  | 0.99 | 0.78 | 1    |
| **uBTMs** | 0.97 | 1     | 0.97 | 0.79 | 0.97 |
| **tSLI**  | 0.99 | 0.97  | 1    | 0.74 | 0.99 |
| **eBTM**  | 0.78 | 0.79  | 0.74 | 1    | 0.78 |
| **sBTM**  | 1    | 0.97  | 0.99 | 0.78 | 1    |

The correlation matrix of the area differences, density correlation, *Deep AI*, and *Resemble.js* recognized the eBTM singularity. The method also identified similar values for the other models (Figure 7a–d).

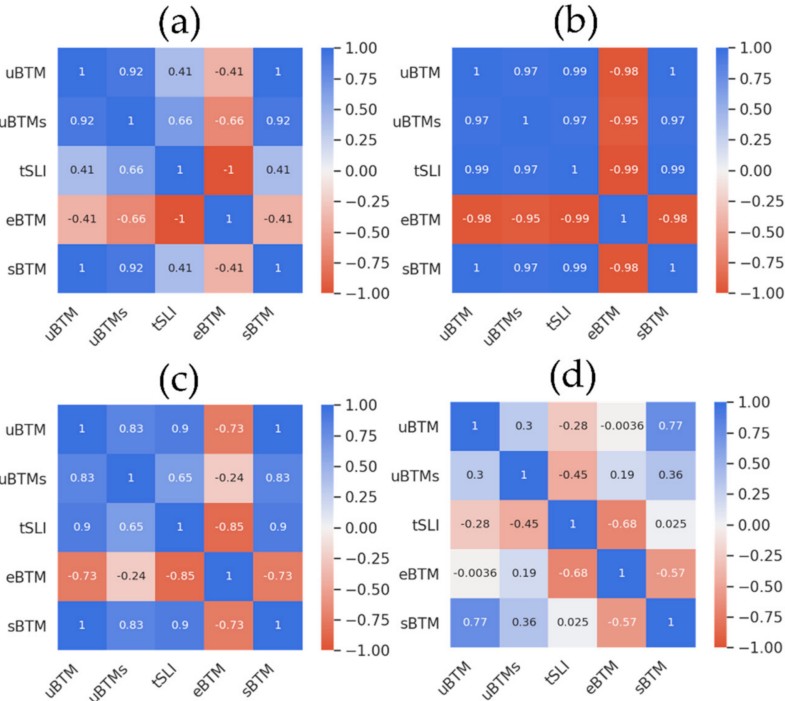

**Figure 7.** Correlation matrix in RSCP. (**a**) Area differences. (**b**) *Kernel Density* correlation matrix. (**c**) *Deep AI* image similarity API. (**d**) *Resemble.js* image similarity API. The values of correlation are included in the squares.

### 3.1.2. Spatial Similarity Analysis: Coast Section

In the coastal stretch, the differences between models are more evident, especially in uBTMs and tSLI models, which show different spatial distributions of density points (Figure 8). The uBTMs smooth process deleted the loose pixels that influenced the *Kernel Density* results (Figure 8e).

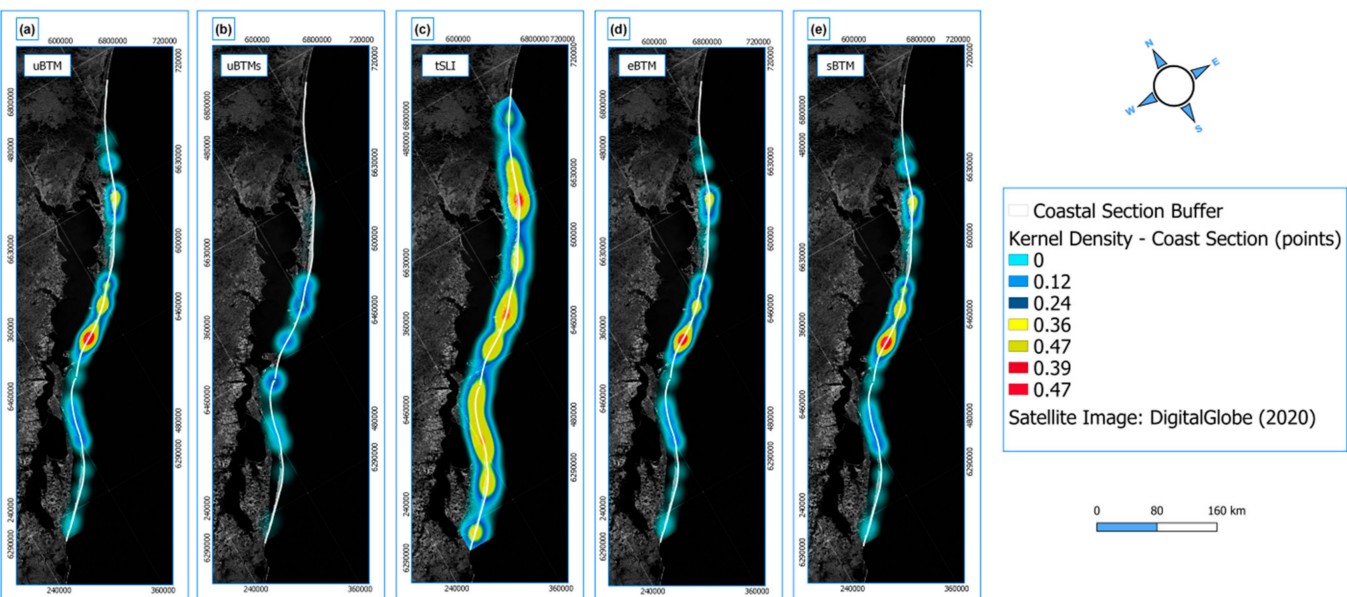

**Figure 8.** Spatial distribution with Kernel Density on coastal section. (**a**) Uncertainty Bathtub model (uBTM). (**b**) Uncertainty Bathtub model smoothed (uBTMs). (**c**) Terrset sea-level impact (tSLI). (**d**) Enhanced Bathtub model (eBTM). (**e**) Simple Bathub model (sBTM).

The uBTM and sBTM has a correlation factor of 1 while with tSLI the correlation value is 0.77. The eBTM presents high correlation values (0.99) with both uBTM and sBTM models (Table 2).

**Table 2.** Correlation matrix of models on the coast section.

|  | **uBTM** | **uBTMs** | **tSLI** | **eBTM** | **sBTM** |
|---|---|---|---|---|---|
| **uBTM** | 1 | 0.75 | 0.77 | 0.99 | 1 |
| **uBTMs** | 0.75 | 1 | 0.70 | 0.73 | 0.75 |
| **tSLI** | 0.77 | 0.70 | 1 | 0.75 | 0.77 |
| **eBTM** | 0.99 | 073 | 0.75 | 1 | 0.99 |
| **sBTM** | 1 | 0.75 | 0.77 | 0.99 | 1 |

The singularities of uBTMs and tSLI on the coastal section are evident on Image Similarity APIs as well. The *Deep AI* and *Resemble.js* also recognized the uBTM, eBTM, and sBTM spatial similarities (Figure 9c,d). This situation makes it clear that the area differences analysis on its own cannot accurately distinguish the spatial distribution between models as reached by the density correlation method and image similarity APIs.

*3.2. BRGM Validation*

The results presented in Table 3 compare the numeric characteristics of the profiles (i.e., berm high, profile length, depth of closure profile, and coastline retreat) obtained by [57] and Bruun rule for GEE model (BRGM). The results of the BRGM with the projection for 2100 (RCP 8.5, 1.21 m) points to a maximum coastline retreat of about 146.6 m close to the south jetty (Profile 1) and a minimum of 78.5 m (Profile 8) (Figure 1a) (Table 3).

According to projections for 2100 under the RCP 8.5 scenario, the coastline might suffer a total retreat of about 100 m (Figure 10).

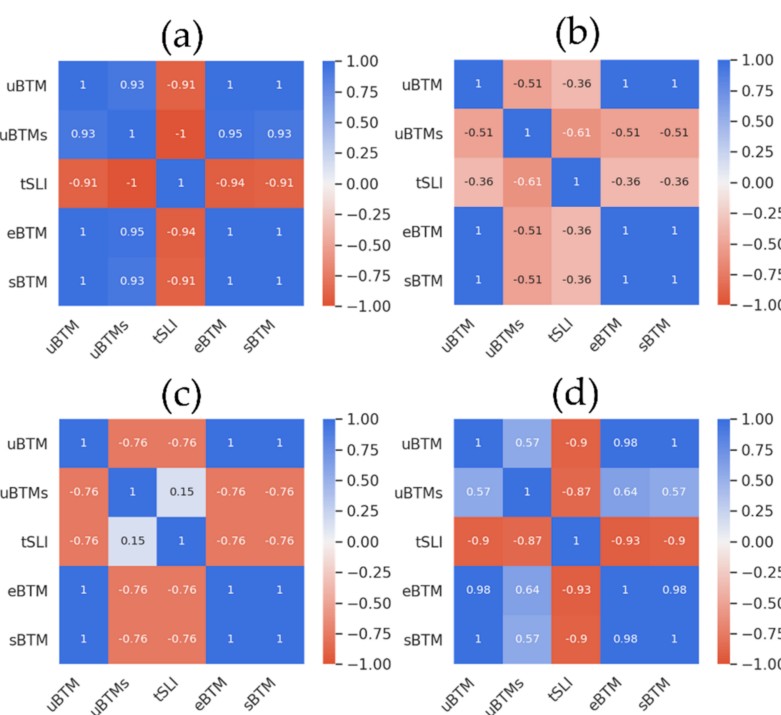

**Figure 9.** Correlation matrix of the coast section. (**a**) Area differences; (**b**) *Kernel Density* correlation matrix. (**c**) *Deep AI* image similarity API. (**d**) *Resemble.js* image similarity. The values of correlation are included in the squares.

**Table 3.** Values of morphological variables obtained by [57] and in the present work. The coastline retreat results (CRR) in both situations are calculated using a sea-level rise (SLR) of 0.50 m. The 2100 RCP 8.5 AIS uses an SLR of 1.21 m. The units of the berm, profile length (W), and depth of closure are in meters (m).

| Profiles | Berm | | W | | Depth of Closure | | CRR | | 2100 |
|---|---|---|---|---|---|---|---|---|---|
| | [57] | BRGM | [57] | BRGM | [57] | BRGM | [57] | BRGM | RCP 8.5 |
| 1 | 5.6 | 5.9 | 2240 | 2291 | −12.1 | −12.8 | 63.3 | 61.2 | 146.6 |
| 2 | 4 | 4.5 | 1440 | 1404 | −12.1 | −11.8 | 44.7 | 43.2 | 105.8 |
| 3 | 1.5 | 0.1 | 1440 | 1412 | −12.1 | −12.3 | 52.9 | 56.8 | 135.5 |
| 4 | 1.9 | 2.1 | 1483 | 1494 | −12.1 | −12.2 | 53.0 | 52.2 | 124.7 |
| 5 | 4 | 4.0 | 1450 | 1514 | −12.1 | −12.2 | 45.0 | 46.7 | 112.0 |
| 6 | 2.4 | 2.6 | 1483 | 1435 | −12.1 | −12.1 | 51.1 | 48.9 | 116.5 |
| 7 | 3.2 | 3.2 | 1333 | 1251 | −12.1 | −11.8 | 43.6 | 41.6 | 99.5 |
| 8 | 8.2 | 9.0 | 1434 | 1387 | −12.1 | −12.1 | 35.3 | 32.9 | 78.5 |
| 9 | 4.8 | 4.2 | 1420 | 1454 | −12.1 | −12.6 | 42.0 | 43.2 | 105.5 |

The nine profiles used to estimate the coastline retreat with BRGM are compared with those of [57]'s study. There is a strong correlation between them (r = 0.97), showing a coefficient of determination ($R^2$) of 0.93, *t*-test (9.49), and *p*-value equals zero, which indicates a very-high coherence between the results of both studies (Figure 11).

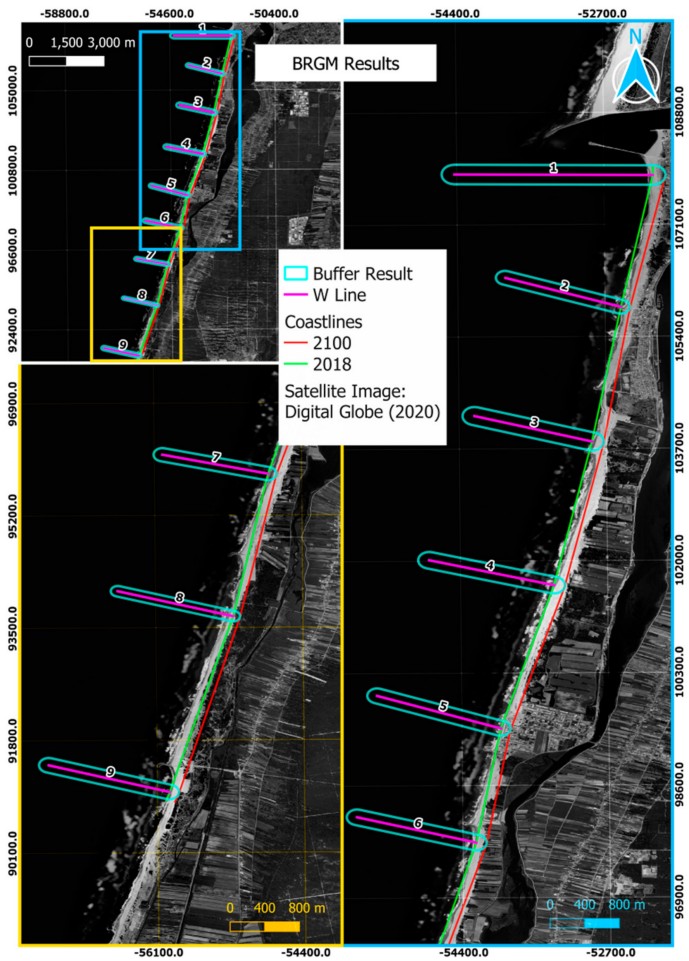

**Figure 10.** Results of Bruun rule by the year 2100 with the projection RCP 8.5 AIS 60 cm (1.21 m). Table 2018. and the red line the expected position in 2100.

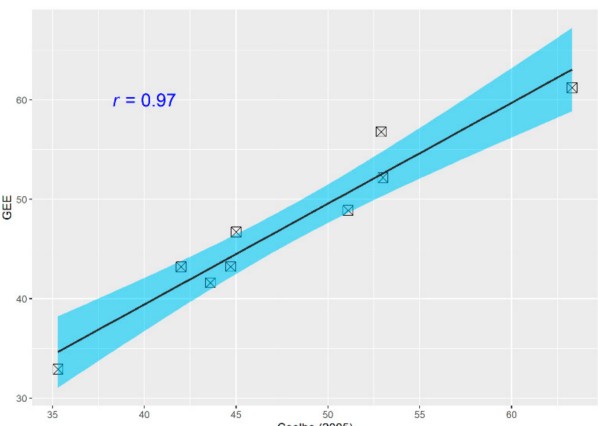

**Figure 11.** Spearman correlation coefficient between coastline retreat along the profiles of GEE model and [57] study ($R^2$ = 0.93, r = 0.97).

## 4. Discussion

The two models presented in this paper (uBTM and BRGM) may correctly operate and were positively validated with other well-known GIS-based models and previous studies. It is suitable to affirm that both models can produce appropriate outcomes according to their objectives.

The uBTM and uBTMs can be defined as being a hybrid between sBTM and tSLI. The uBTM has the advantage of representing the sBTM in a probabilistic form related to uncertainties and reducing the computational complexity. The main difference between tSLI and the remaining models is due to the PCLASS algorithm used by Terrset that operates with a different reclassification, which calculates the area under a normal curve defined by the threshold value using the uncertainties as standard deviation [19]. The *Circle Kernel* filter applied to the uBTMs model reduced the scarce pixels and improved the delimitation of the Rio Grande do Sul coastal plain (RSCP) coastline contour. Furthermore, the pixels removed reduced the correlation with the sBTM in the coast stretch but when the total of the RSCP is considered the uBTMs presents 0.97 of correlation. Zones of high vulnerability like salt marshes [61], low altitude and flat areas [62], and places prior recognized as a priority for coastal management [63,64] were coherently recognized by all the models as areas with a high risk in flood situations caused by sea-level rise. Furthermore, a recent study [65] performed at Cassino beach (Figure 1a) with the bathtub model disclosed areas quite similar than the obtained in the present study if the uBTM doesn't consider the uncertainties. The uBTM and uBTMs works data with uncertainties, due to this, if the DEM/DTM is obtained through from unoccupied aerial vehicle (UAV) and Light Detection and Ranging (LiDAR) with high vertical resolution accuracy the changes due to uncertainties will not be visible. For the aforementioned reason, the uBtM and uBTMs are recommended only for medium or low-resolution data, such as AW3D30 [66], SRTM [67], CoastalDEM [53], or other products that requires an uncertainty analysis.

Combining the correlation matrix results with the Kernel Density and the image similarities APIs, it is possible to recognize the spatial patterns of eBTM and tSLI and similarities to other models in general. The inclusion of artificial intelligence as a tool to compare images and recognize spatial designs and trends can bring useful algorithms to the existing GIS software available nowadays. Additionally, it is essential to highlight that the eBTM results reveal the hydrological connectivity of the lagoons with success. In this case, as in uBTMs, the low correlations do not necessarily determine inferior quality results.

Regarding the BRGM, the high correlation of 0.97 with [57] results proves that this model can perform the Bruun rule on Google Earth Engine with success. The correlation value is not one since challeging to set the same values of profile length, closure of depth, and berm heigh using different topo-bathymetric data utilized by [57].

Recently, [68] published a study using the Bruun rule, and [69] criticized the authors for using the Bruun rule without considering the offshore sediment transport. However, a recent study used Bruun rule to evaluate the sea-level rise impact on tourism [70]. The study was conducted in São Jacinto beach, north of Aveiro harbour (Figure 1b), and the results for 2050 are similar with the half-value for 2100 prediction of BRGM. Overall, it is always essential to remember that the analysis may not be conclusive because the original equation ignores some factors such as the overwash events and changes in sediment supply due to natural and antropogenic operations (such as dredging and nourishment operations [71] that might continue being performed in the area). Nevertheless, the implementation of the original Bruun rule in GEE can turn easier to apply, helping to understand the formula better and providing a new environment in GIS that can encourage the creation of more realistic modifications of the Bruun rule itself.

The models should be used with caution due to their inherent simplicity it is sufficient to conclude about sea-level impact by using these analyses alone. Though, these same simplicity of operations of both models avoid GEE limitation to handle functions that involve long-running iterative processes, such as finite element analysis, agent-based models, and machine learning models [12]. Furthermore, these methodologies take few minutes to run and are useful for an initial assessment of sea-level impact enhancing coastal management, to be supported by more detailed studies combined with other models and including more variables. The models eBTM and tSLI also took few minutes to run,

otherwise, the uBTM does not need any software installations and can be run directly in the web browser.

The future work consist of implementing the models as GEE application, integrating the eBTM in uBTM, and performing the validation of the other Bruun rule variations of [55,56] equations already included in the code. Both methods, uBTM and BRGM can be found to download in the references [18,20].

## 5. Conclusions

This work presents and validates two models for the assessment of sea-level rise created on Google Earth Engine (GEE). The GEE has shown to be a useful analytical platform to develop models that can be performed in different studies of coastal dynamics. Both models bring advantages for the coastal management in GEE cloud-based platform. Other highlight of this study includes the application of artificial intelligence which was tested with success to validate the spatial distribution.The Uncertainty Bathtub Model (uBTM) reveals high similarities and correlations with tested models. This proved uBTM as a reasonable option to represent the impact of the sea-level flood. Likewise, the Bruun Rule for GEE Model (BRGM) validation allowed a high degree of confidence that guarantee the model is well adjusted. Due to GEE's characteristics, this model can now run efficiently in a cloud-based GIS environment, promoting Bruun Rule improvements by calibrations, modifications, and enhancing its base formulation. The uBTM and BRGM codes are in open access for the scientific community [18,20], and thus, they can make updates and adapt the code to its applications and scientific investigations.

**Author Contributions:** Conceptualization, L.T.d.L.; data curation, J.F.G.; formal analysis, L.T.d.L.; investigation, S.F.-F.; methodology, L.T.d.L.; software, L.T.d.L.; supervision, C.B.; validation, L.T.d.L. and S.F.-F.; visualization, L.T.d.L.; writing—original draft, L.T.d.L. and S.F.-F.; writing—review & editing, S.F.-F., J.F.G., L.M.F. and C.B. All authors have read and agreed to the published version of the manuscript.

**Funding:** L.T. de Lima is grateful to the Brazilian National Council for Scientific and Technological Development (CNPq) for the CSF's (Ciência sem Fronteiras) program doctoral fellowship granted (249636/2013-1) J.F. Gonçalves is financially supported by the Fundação para a Ciência e a Tecnologia (FCT) through contract number: CEECIND/02331/2017/CP1423/CT0012. J.F. Gonçalves developed this work within the scope of the project proMetheus—Research Unit on Materials, Energy and Environment for Sustainability, FCT Ref. UID/05975/2020, financed by national funds through the FCT/MCTES. Thanks are due to FCT/MCTES for the financial support to CESAM (UIDP/50017/2020+UIDB/50017/2020), through national funds.

**Data Availability Statement:** The following are available online at http://doi.org/10.5281/zenodo.4664207, Including DEM/DTM input data of RSCP and Barra -Poço da Cruz coastal strecht; BRGM results for 2100 (RCP 8.5) in Barra-Poço da Cruz coastal strecht; Results of uBTM, uBTMs, eBTM, tSLI, and sBTM in RSCP for 2100 (RCP 8.5).

**Acknowledgments:** The authors gratefully acknowledge editor Monica Timar for her helpful assistance and anonymous reviewers for their insightful comments.

**Conflicts of Interest:** The authors declare no conflict of interest.

## Appendix A

Graphical model for the validation process in ArcGIS (version 10.6). Blue circles are the inputs; yellow rectangles are the tools; green circles are the outputs.

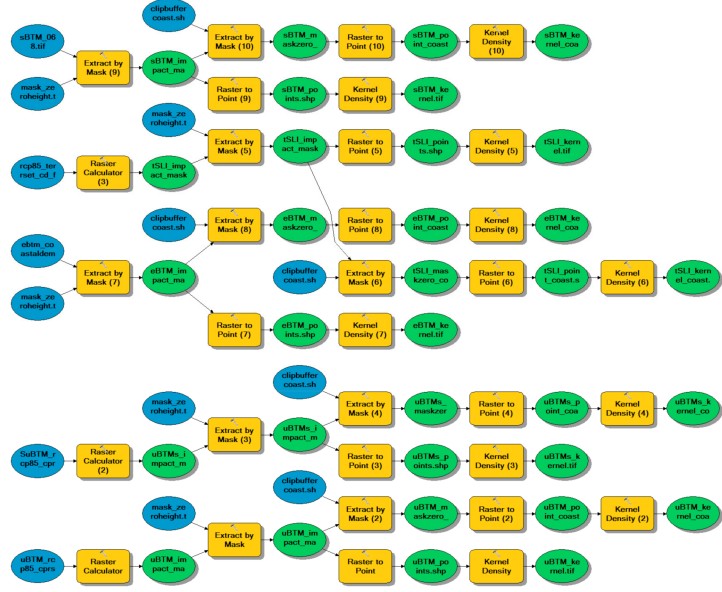

**Appendix B**

Other equations included in the BRGM:
The Bruun equation can be re-written as:

$$R = S/(\tan(\beta))$$

where β is the average beach slope between the berm ridge or dune foot and the depth of closure.

Ref. [55] to extend landwards transport that adds the variant $V_D$ that represents the deposited sand volume;

$$R = S\,(W + V_D/S)/(h + B)$$

Additional modifications in [56] add contributions of the cross and longshore sedimentary processes, and the sediment budget was also included.

$$R = S/\tan(\beta) + \text{fcross} + \text{flong}$$

*fcross* and *flong* are the contributions of processes causing losses or gains of sediments in the active beach profile.

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
