# Peer review of "Development of Tools for Coastal Management in Google Earth Engine: Uncertainty Bathtub Model and Bruun Rule"

_remotesensing, doi:10.3390/rs13081424_

Round 1

Reviewer 1 Report

It is a usefull work for practical application of GEE in coastal management. Authors developed two models on GEE and validated them in two sites. The whole manuscript is well written. General comments are as follows:
1.Why not validate the two models in both sites? According to the paper: The RSCP in the south of Brazil was chosen to validate the Uncertainty Bathtub Model (uBTM), and the second site considered, in order to validate the Bruun Rule for GEE Model.
2.The costs between offline GIS models and online GEE models may be compared, including the input preparation and running time.
3. Is it possible to compare the coastline result to CoastSat?
Specific comments include:
Figure 1, What is Bing satellite?
Figure 2, The full name of USP and VE should be given.
Figure 3, What does the 1234567890 means in the the numerical result.
Table 1, Besides the good values (0.97), the low R of 0.74 should be mentioned in the main texts.
Table 3, The format of table 3 seems different from other tables.  There are only results of Closure depth and 2100 for BRGM, which is not consistent to the title (comparison).   

Author Response

Dear Reviewer,

We acknowledge your insightful comments that have contributed to improve the original manuscript. Below we respond to your comments and we inform that all changes were done in the original manuscript using the option track-changes. Furthermore, English language and style were revised.

Point 1: Why not validate the two models in both sites? According to the paper: The RSCP in the south of Brazil was chosen to validate the Uncertainty Bathtub Model (uBTM), and the second site considered, in order to validate the Bruun Rule for GEE Model.

Response 1: The existence and/or availability of field data with good quality to perform the validation of can be a challenge task. Fortunately, in Portugal, University of Aveiro has been monitoring the study area for more than two decades and there is also “Coastal Monitoring Program of Continental Portugal - COSMO” [59] that allows having access to free high-quality topo-batymetric data to validate Bruun Rule for GEE Model (BRGM). Unfortunately, in Brazil, there was not access to all data needed in order to validate BRGM. Therefore, BRGM was validated in Aveiro (Portugal) and uncertainty Bathtub Model (uBTM) was validated in Brazil because Rio Grande do Sul Coastal Plain (RSCP) is well known by the first and fourth authors and thus, the gaps in the uBTM results could be easily identified. The reason to validate these models in these study sites was added in lines 176-177, 180-182, and 203-205.  

Point 2: The costs between offline GIS models and online GEE models may be compared, including the input preparation and running time.

Response 2: The information about offline and online GIS models was included in Discussion section in the line 564. The differences of running time of all models in this study case are insignificant in all the models; this information was added in the line 563.

Point 3:. Is it possible to compare the coastline result to CoastSat?

Response 3: The CoastSat is a tool that helps to extract the shoreline/coastline position in Landsat and Sentinel Satellite Images, but it does not estimate the shoreline/coastline position in the future as Bruun Rule for GEE Model does.

Point 4: Figure 1, What is Bing satellite?

Response 4: The basemap used to create Figure 1 was obtained from Bing Maps service whose satellite images are provided by DigitalGlobe. The correct information was added in Figure 1.

Point 5: Figure 2, The full name of USP and VE should be given.

Response 5: Figure 2 was modified according to Reviewer 2’ suggestions.

Point 6: Figure 3, What does the 12345 and 67890 means in the numerical result.

Response 6: These numbers are only a visual representation of a numerical result. Figure 3 was modified according to Reviewer 2’ suggestions.

Point 7: Table 1, Besides the good values (0.97), the low R of 0.74 should be mentioned in the main texts.

Response 7: The value 0.74 is related with eBTM which takes into account the water connection and excludes the lagoons. This information was added in lines 424-426.  

Point 8: Table 3, The format of table 3 seems different from other tables.  There are only results of Closure depth and 2100 for BRGM, which is not consistent to the title (comparison).

Response 8: The format of Table 3 was reviewed and the title was modified. The Closure depth obtained by [58] was included. Furthermore, more information about the values was added in Discussion section in lines 538-540.

Reviewer 2 Report

My comments are included in the attached pdf file.

Author Response

Dear Reviewer,

We acknowledge your insightful comments that have contributed to improve the original manuscript. Below we respond to your comments and we inform that all changes were done in the original manuscript using the option track-changes. Furthermore, English language and style were revised.

Point 1: Line 16: Free and Open Source Models. Why capital letters?

Response 1: The capital letters were removed.

Point 2: It is a personal opinion as regular reader of this journal. This Introduction is too long and subsections are not needed. So, I disagree with the length of the content and with its structure. Therefore, I suggest this section should be fully rewritten. Actually, authors only have to reorganize the content.

Response 2: This section was rewritten according to the suggestions.

Point 3: Line 43: Please, remove ‘Total’

Response 3: It was removed.

Point 4: Lines 31-59: First of all, congratulations to authors because they write really well. It is a pleasure to read an article in which everything is understandable and, in addition, the style is rigorous and refined. Secondly, the information expressed in these first three paragraphs are very interesting and useful. Nonetheless, taking into account the scope of the RS journal I think authors should reduce substantially the content of these three paragraphs. This is my personal opinion.

Response 4: Thank you for your kind words. The content of paragraphs was reduced. 

Point 5: 1.4. Study Area: This subsection should be the first one of the M&M section. Perhaps, I am very traditional regarding the structure of research article but I often think the conventional structure is the best for the reader.

Response 5: This subsection was moved to Material and Methods section.

Point 6: Lines 113-114: Why did you choice these study sites?

Response 6: The reason to choose these study sites was included in the lines 176-177, 180-182, and 203-205.

Point 7: Figure 2: This figure does not provide anything relevant to understand the workflow of this research. I think it is much better to draw illustrative schemes more specific in some subsections that this so generic here. Opinion as reader.

Response 7: Figure 2 was redesigned to add more information in order to improve the explanation of the model.

Point 8: 2.2 uBTM Validation: This subsection is a little bit long. I think authors could make an effort to be more concise, i.e. fewer lines and paragraphs. There are some descriptions regarding ArcMap that are too basic to be described in the text.

Response 8: The length of the subsection was reduced and the ArcMap information was removed.

Point 9: Figure 3: I am very respectful with the way of drawing that authors have chosen. In my opinion, you are wasting a lot of space by using this kind of figures. If I were you I would do illustrations with more soverty, less visual but robust including relevant information.

Response 9: Figure 3 was redesigned.

Point 10: Figure 5: Please improve the visual quality of this figure.

Response 10: The quality of Figure 5 was improved.

Point 11: Results. Sections, subsections and sub-subsections is much more than enough to show your results. I think a fourth level of data disaggregation is too much. Please think also about the readiness of your manuscript. We are scientists but also readers.

Response 11: We completely agree, and the fourth level of sections was removed.

Point 12: Figures 7 and 9: This information should not be shown as a figure. It is much better a table. I have done once and I regretted because information is better than the way of visualization.

Response 12: Figures were improved adding the correlation values to enhance the understanding. Anyway, if you consider that the improvement is not enough, they can be converted in tables.

Point 13: Line 354: 3.1.2.2 Coast section: 4th level is too much (hyper-structuration of the text).

Response 13: This subsection was removed.

Point 14: Discussion: Contrariwise, this section is too short in comparison with the length of the rest of the sections.

Response 14: Discussion section was improved adding advantages and disadvantages of the models and comparing results with previous works in the study sites. This information was included in the lines 520-228, 538-540, 543-546, 458-560, 557-560, 563-568.

Point 15: Conclusions: I think you should show this section as a single paragraph.

Response 15: The conclusion was reformulated in a single paragraph.

Reviewer 3 Report

 Development of tools for coastal management in Google Earth Engine: Uncertainty Bathtub Model and Bruun Rule

Dear Editor

I have read above article very carefully. This is an interesting work. My suggestion is acceptance after minor revision.

Please work on language manuscript and edit by a native.

 Abstract

I cant see any quantitative results and even a conclusion in this part.

  1. Introduction

What is innovative of this study? Any details on this subject?

Please concentrate on study area and then you could discuss on GEE and other relations. Please replace these parts.

1.4. Study Sites

Please add coordinate systems of the study sites.

Any details on climate features?

  1. Discussion

This part needs to empower by some previous publications. Please discuss on advantages, disadvantages and future tasks.

I think authors have to share their code in GEE.

Author Response

Dear Reviewer,

We acknowledge your insightful comments that have contributed to improve the original manuscript. Below we respond to your comments and we inform that all changes were done in the original manuscript using the option track-changes. Furthermore, English language and style were revised.

Point 1: Please work on language manuscript and edit by a native.

Response 1: A qualified person reviewed the language manuscript.

Point 2: Abstract: I can’t see any quantitative results and even a conclusion in this part.

Response 2: The quantitative result is the high correlation between the created model and other models and studies. The sentence can be found in lines 21-22, and 24-25.

Point 3: Introduction: What is innovative of this study? Any details on this subject? Please concentrate on study area and then you could discuss on GEE and other relations. Please replace these parts.

Response 3: This work shows the potential of GEE to support Uncertainty Bathtub model and Bruun Rule model for the first time. Additionally, these models are validated as example in two places which are exposed to coastal hazards (erosion and flooding) but they might be validated in other places exposed to the same hazards where good quality field data would be available. According to the journal, study area should be included in Material and methods section instead in Introduction section. Therefore, this subsection was moved to Material and methods section.

Point 4: 1.4. Study Sites: Please add coordinate systems of the study sites. Any details on climate features?

Response 4: The coordinate systems were added in lines 28 and 29. Climate features are not relevant in the performed work and thus, this information was not included.

Point 5: Discussion: This part needs to empower by some previous publications. Please discuss on advantages, disadvantages and future tasks.

Response 5: Previous studies were included in Discussion in lines 520-522, and 543-545. Furthermore, advantages, disadvantages and future task were added in lines 563-568, and 574-576.

Point 6: I think authors have to share their code in GEE.

Response 6: The codes are available in Zenodo [references 18 and 20] and this was mentioned in lines 568, and 585.
